# Quality and Security Signals in AI-Generated Python Refactoring Pull Requests

### Mohamed Almukhtar
almukhtr@umich.edu
University of Michigan-Flint
USA

### Anwar Ghammam
aghammam@umich.edu
University of Michigan-Dearborn
USA

### Hua Ming
huaming@umich.edu
University of Michigan-Flint
USA

## Abstract

As AI agents increasingly contribute to code development and maintenance, there is still limited empirical evidence on the quality and risk characteristics of their changes in real-world projects, particularly for refactoring-oriented contributions. It remains unclear how agent-authored refactoring edits affect maintainability, code quality, and security once merged into GitHub repositories. To address this gap, we conduct an empirical study of Python refactoring pull requests (PRs) from the AIDev dataset. We analyze agentic refactoring PRs using PyQu, an ML-based quality assessment tool for Python, to quantify changes across five quality attributes, and we complement PyQu with domain-independent static analysis (Pylint and Bandit) to measure code-quality and security issues before and after each change.

Our results show that, on average, agentic commits improve a quality attribute in 22.5% of the studied changes, with usability improving most frequently (36.5%). At the same time, 24.17% of modified files introduce new Pylint issues—predominantly convention-level violations such as long lines—while 4.7% introduce new Bandit findings. From the observed diffs, we derive a taxonomy of 24 recurring change operations and map them to the lint and security findings they most commonly affect. Despite these mixed outcomes, developer acceptance is high: 73.5% of the analyzed PRs are merged, including cases that introduce new lint or security findings, often alongside the removal of existing issues. Overall, these findings highlight both the promise and current limitations of agentic refactoring, and motivate stronger tool-in-the-loop quality and security gating for AI-driven development workflows.

## Keywords

Agentic AI, Code Generation, Code Quality, Software Security

**ACM Reference Format:**
Mohamed Almukhtar, Anwar Ghammam, and Hua Ming. 2026. Quality and Security Signals in AI-Generated Python Refactoring Pull Requests. In *Proceedings of 3rd ACM International Conference on AI-powered Software (AIware 2026)*. ACM, New York, NY, USA, 10 pages. https://doi.org/XXXXXXX.XXXXXXX

## 1 Introduction

Modern Software Engineering is undergoing a shift as autonomous AI agents increasingly contribute directly to code development work [15, 17]. Instead of acting only as code-completion tools, newer agentic systems behave more like teammates: they plan tasks, navigate repositories, run multi-step actions, and produce code changes with limited human supervision [14, 15, 24]. Recent advances show that these agents can perform development tasks such as implementing features, refactoring code, fixing bugs, updating documentation, and opening Pull Requests (PRs) in real repositories [16, 17, 32].

This growing autonomy creates new opportunities, but also raises basic questions. AI-generated PRs can speed up and streamline development, yet their increasing presence in real-world code review workflows makes it important to understand the quality, reliability, and safety of the changes they introduce [16, 21, 26, 32]. Even with rapid progress in agent-based development, systematic empirical evidence remains limited on how these agents affect multidimensional code quality in Python refactoring PRs [16, 32], what risks they introduce or remove, and how their PRs fit into real review and merge workflows.

To address this gap, we leverage the AIDev dataset [18] to study the multidimensional quality impact of AI-generated code changes, quantify the prevalence of static-analysis and security signals, and characterize recurring patterns of strengths and weaknesses across autonomous coding agents. We scope this study to Python refactoring PRs. We focus on Python because recent empirical evidence identifies it as a dominant default language in AI-assisted code generation. Large Language Models (LLMs) show a strong default preference for Python in language-agnostic programming tasks and continue to favor it even when alternative languages may be equally suitable [29]. This prevalence makes Python an especially relevant setting for examining AI-mediated development practices.

We further restrict our analysis to refactoring PRs to reduce confounding effects from feature additions and bug fixes. By prioritizing behavior-preserving transformations, which are generally intended to restructure code without changing external behavior, we can more directly attribute observed changes (whether improvements or regressions) in software quality attributes and static-analysis outcomes to structural code modifications rather than to newly introduced functionality. This design choice strengthens the internal validity of our empirical analysis by isolating the impact of code restructuring activities.

To enable this study, we utilized PyQu [8], a quality-enhancement detection tool that combines low-level code metrics with machine-learned models to predict whether a change affects any of five quality attributes (QAs), as detailed in subsection 2.2. In addition,

we use Pylint [19] to capture code-quality issues and Bandit [1] to capture security-related issues. We complement these automated analyses with statistical testing and targeted manual inspection to validate interpretations.

In this context, we answer the following research questions:

**RQ1:** **Do agentic refactoring PRs improve Python software quality?**
This RQ examines how often AI-generated refactorings in Python code improve each of the five QAs measured by PyQu [8]. Overall, we observe an average quality enhancement rate of 22.5% of the commits across the five QAs, with usability emerging as the most frequently improved QA (273/747 commits, 36.5%).

**RQ2:** **How often do agentic refactoring PRs introduce or remove code and security issues in Python software?**
This RQ investigates code-quality and security issues in agent-generated refactoring PRs by tracking which issues are introduced or removed, and by examining the code-change operations that lead to issue removals. We use Pylint [19] to capture code-quality issues and Bandit [1] to capture security issues. Overall, 24.17% (611/2528 files) of analyzed files introduced new Pylint issues (predominantly stylistic), while 4.7% (119/2528 files) introduced new Bandit issues, which were mainly risky coding practices rather than high-severity vulnerabilities.

**RQ3:** **To what extent are agentic refactoring PRs accepted by developers?**
This RQ examines whether developers accept agent-generated refactoring PRs by merging them, and characterizes non-merge outcomes (i.e., PRs closed without merge). Overall, we observe a high acceptance rate: 73.5% (322/438) of agentic refactoring PRs were merged, while 84 were closed without merge, often with no explicit rationale recorded in the PR discussion. Notably, acceptance frequently occurred despite regressions in automated checks: 129 merged PRs introduced new Pylint issues, and 27 merged PRs introduced new Bandit issues.

This paper makes the following contributions:

- **A multi-tool measurement framework for agentic refactoring impact.** We operationalize refactoring impact using PyQu (five quality attributes), Pylint (code-quality issues), and Bandit (security issues), enabling complementary views of quality and security changes.
- **Empirical evidence on issue churn in agentic refactoring.** We quantify how often agentic refactorings introduce, remove, or leave unchanged code-quality and security issues at the file level, revealing substantial churn in Pylint signals and comparatively stable Bandit signals.
- **A taxonomy of change operations linked to static-analysis issues.** We derive a categorized set of recurring refactoring operations (e.g., security remediation, maintainability refactoring) and map them to the Pylint/Bandit identifiers they most frequently affect.
- **An analysis of developer acceptance and enforcement practices.** We study merge decisions and show that a large

fraction of agentic refactoring PRs are accepted, including cases that introduce new lint and security issues.
- **Replication package for transparency and reuse.** We release the extracted measurements, and analysis scripts to facilitate replication and future studies on agentic code changes [6].

## 2 Methodology

### 2.1 Dataset

We base our analysis on the AIDev dataset [18], a large-scale dataset of around 933k agentic PRs spanning real-world GitHub repositories and generated by five AI agents: OpenAI Codex, Devin, GitHub Copilot, Cursor, and Claude Code. From repositories with more than 100 stars, it includes 33,596 PRs and 39,122 agent-generated commits, spanning 12 task types: feat, fix, docs, test, refactor, chore, build, ci, perf, style, other, and revert.

To target popular repositories, we first filtered projects with more than 100 stars. We then refined the dataset to Python projects by identifying repositories whose primary language is Python and retaining only the corresponding PRs. Given our emphasis on quality-oriented modifications, we limited the dataset to PRs labeled as `refactor`. Such PRs are well aligned with our objectives, as they are intended to improve code structure and maintainability—rather than introduce new features or fix defects—thereby reducing the likelihood of false positives in our analysis. This filtration step yielded a total of 438 refactoring PRs, distributed across the AI agents as follows: 321 generated by OpenAI Codex, 42 by Copilot, 38 by Devin, 31 by Cursor, and 6 by Claude Code. Since our analysis tools (PyQu, Pylint and Bandit) operate at the commit level, we extracted all commits associated with the selected PRs, yielding a total of 1,171 commits. After additional cleaning, we removed 285 commits that did not contain any Python files and 16 duplicate commits, resulting in a final dataset of 870 commits.

### 2.2 RQ1: Agentic-PRs Quality Assessment

To evaluate the quality of the agentic refactoring changes in Python, we used PyQu [8], an ML-based quality assessment tool for Python. PyQu was previously evaluated on 500 commits that were manually labeled by three authors [8] and achieved an average accuracy of 0.84 and F1 score of 0.85. Its empirically validated performance and metric-driven design make it well-suited for systematically identifying quality-oriented changes in our dataset.

PyQu operates in two stages. First, it computes a set of low-level code quality metrics (e.g., cyclomatic complexity, documentation density, coupling, and cohesion) for each commit. Second, it applies machine-learned classifiers to map these metrics to five high-level quality attributes: understandability, reliability, maintainability, modularity, and usability. Although the classifiers were trained on commits from Python ML projects, the underlying metrics are domain-agnostic and widely used in software quality research. PyQu labels a QA as "Enhanced" when it improves after a commit changes and as "Not Enhanced" when the quality stays the same or decreases.

Following prior work on cross-project prediction and model transfer [34], we use PyQu as a consistent and comparative signal

of quality change across AI-generated pull requests, while interpreting its outputs as relative indicators rather than absolute quality judgments. We therefore interpret PyQu's outputs (low-level code quality metrics) as indicators of how an agentic PR affects structural properties of the code. After running the tool on our list of 870 commits extracted from subsection 2.1, only 747 commits produced results. The remaining commits could not be processed because they contained Python 2 files that could not be successfully converted to Python 3.

For the commits labeled as `Not Enhanced` by PyQu, we conducted a deeper analysis to better understand the underlying reasons for this classification. Two authors with experience in software engineering, software maintenance, and evolution manually and independently inspected a random subset of 100 commits from the `Not Enhanced` group. For each commit, we reviewed the change diff and commit/PR context to (i) characterize the type of operation performed (e.g., renaming, cleanup, restructuring) and (ii) judge whether the change plausibly improves any of the PyQu-supported attributes or their underlying metrics. Each author recorded a label for the change type along with brief notes explaining their reasoning. We then measured inter-rater agreement using Cohen's kappa, obtaining $\kappa = 0.81$, which indicates strong agreement [31]. Finally, the two authors met to discuss disagreements and produced a single adjudicated label for each commit.

*2.2.1 Statistical Analysis.* We evaluate whether commits labeled `Enhanced` vs. `Not Enhanced` differ systematically in their low-level metric deltas. To move beyond descriptive summaries (e.g., mean deltas) and provide stronger evidence that PyQu labels correspond to measurable quality shifts, we test whether the metric-delta distributions differ between the two groups. For each quality attribute, we conduct two complementary analyses: (i) a *confirmatory* analysis restricted to the metric deltas that PyQu associates with that QA [8], and (ii) an *exploratory* analysis over all deltas in our extracted PyQu output. For each metric, we compare the `Enhanced` and `Not Enhanced` groups using a two-sided Mann–Whitney U test (to test for a distributional shift between groups) and we report Cliff's $\delta$ as a non-parametric effect size (quantifying the magnitude and direction of the difference), along with bootstrap 95% confidence intervals [7, 20]. We control for multiple comparisons using the Benjamini–Hochberg false discovery rate (FDR) procedure, which adjusts $p$-values to control the expected proportion of false positives among the results declared significant. We apply BH within each attribute's mapped metric set for the confirmatory analysis, and across all tested metrics for the exploratory analysis [9].

## 2.3 RQ2: Agentic Code Defects and Security issues

To answer RQ2, we used the 870 extracted commits described in subsection 2.1. For each commit and its parent, we analyzed only the modified Python files and applied Pylint and Bandit. In total, the subject commits contain 4,922 files, of which 2,722 were Python files. Pylint [19] is a Python static analyzer that reports five message types (fatal, error, warning, convention, and refactor), while Bandit [1] is a static analyzer for identifying security issues in Python code; each Bandit finding is mapped to a corresponding Common Weakness Enumeration (CWE) entry [3].

Following prior work [28], we apply filtering steps to Pylint outputs. Specifically, we exclude purely stylistic messages (e.g., invalid naming, missing final newline, trailing blank lines, whitespace-only issues, and indentation style warnings such as W0311 (`bad-indentation`)), since these primarily reflect formatting preferences rather than substantive code-quality concerns. We also exclude import-related warnings (e.g., unresolved imports), as Pylint can be unreliable for determining correct import usage across diverse project configurations and environments [30]. Finally, we discard fatal messages. For Bandit, because finding B101 (`assert_used`) is common in test code and can inflate security counts without reflecting production risk, we exclude test files from Bandit analyses. We identify test files using path-based heuristics (e.g., `test/`, `tests/`, `_test.py`, `test_*.py`) and apply the same filter consistently to parent and child versions when computing introduced/removed findings. The raw outputs were parsed into structured records: for Pylint, each line was tokenized as (`path`, `code`, `message`, `location`, `line content`), and for Bandit as (`path`, `test_id`, `message`, `severity/confidence`, `line content`). For each changed file, we parsed the parent and child Pylint and Bandit reports and constructed sets of message identities. We then computed introduced messages (present in the child but not the parent) and removed messages (present in the parent but not the child), and defined $net = |introduced| - |removed|$. Files were labeled as introduced if net>0, removed if net<0, and neutral otherwise. Overall, 2,528 files produced usable paired outputs; the remaining files were excluded because they were newly added or deleted (making comparison impossible) or because the analysis tools failed on one of the versions.

*2.3.1 Manual analysis of high-impact issue removals.* To better understand how agentic commits eliminate lint and security issues, we conducted a focused manual analysis of cases with the largest reductions in lint and security-reported issues. Specifically, we selected 100 high-impact cases from the commits that removed the highest counts of issues, with 70 cases drawn from Pylint removals and 30 from Bandit removals (Bandit removals were substantially less frequent in the dataset). This sampling is intentional: our goal is to capture the dominant operations used in commits where issue removal is most pronounced, rather than to estimate prevalence across all commits. Two authors independently examined each case by reading (i) the reported message (Pylint code/category or Bandit test/CWE), (ii) the parent-version code region associated with the finding, and (iii) the corresponding change in the subject commit. For each case, the authors assigned an operation label describing the primary mechanism by which the finding disappeared (e.g., direct fix such as replacing APIs, refactoring-induced restructuring, code deletion, or code relocation). We measured inter-rater agreement using Cohen's kappa and obtained $\kappa = 0.82$, indicating strong agreement. Remaining disagreements were resolved through discussion to produce a single adjudicated label per case. After that the two authors worked in categorizing the code changes based on the change characteristics.

## 2.4 RQ3: Agentic-PRs Status Analysis

To answer RQ3, we analyzed the 438 Python refactoring PRs described in subsection 2.1 and linked each PR to the code-quality

and security deltas defined in subsection 2.3. We first summarized PR outcomes (merged vs. closed). For merged and closed PRs, we then tested whether merge decisions were associated with (i) introducing new issues and/or (ii) removing pre-existing issues, using *pylint* for code-quality issues and *Bandit* for security issues. To mitigate selection effects, we report results separately for PRs with available tool artifacts and explicitly account for PRs without usable tool output. Finally, for PRs that were closed without merge, we qualitatively inspected PR timelines (commits and discussion) to extract stated closure rationales.

## 3 Results

### 3.1 RQ1: Agentic-PRs Quality Assessment

Figure 1 visualizes how often each agent produces commits labeled as Enhanced by PyQu for each quality attribute. Rows correspond to agents and columns to QA. Cell color encodes the enhancement rate, computed as *Enhanced/(Enhanced+Not Enhanced)*, and each cell is annotated with the raw counts (enhanced/total) to make sample size explicit.

Overall, enhancement is most common for Usability (273/747 commits, 36.5%), followed by Reliability (206/747, 27.6%) and Understandability (179/747, 24.0%). In contrast, Maintainability enhancements are less frequent (111/747, 14.9%), and Modularity is the hardest to improve, with only 71/747 (9.5%) labeled as enhanced.

Looking across agents, the enhancement rates are broadly similar in aggregate, but there are clear attribute-specific differences. Cursor stands out on Understandability (34/99, 34.3%) and also shows a high Usability rate (40/99, 40.4%). Devin and OpenAI_Codex are comparatively stronger on Reliability (43/145, 29.7% and 103/354, 29.1%, respectively). For Maintainability and Modularity, all agents remain low. Finally, Claude_Code has very few samples (only 7 per QA), so its percentages may not be informative.

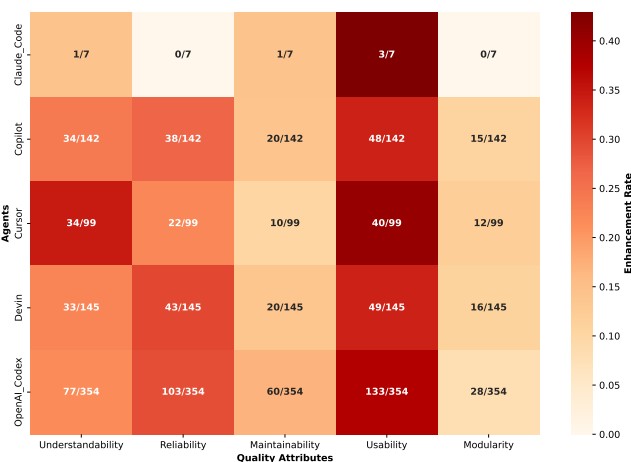

**Figure 1: Enhancement Rates by Agent and Quality Attribute.**

***Why refactoring PRs are not labeled as Enhanced?*** To contextualize the large fraction of Not Enhanced labels, two authors

manually reviewed a subset of the commits as detailed in subsection 2.2. Through this analysis, we observed that many changes were small and mechanical (e.g., renaming variables/modules or minor type edits), which can be beneficial but may not shift PyQu's attribute signals enough to cross the enhancement threshold. We also observed basic cleanup (e.g., removing unused imports, extracting variables), which more often aligns with measurable quality improvements. In many cases, these improvements were not substantial enough to change the overall label.

***Confirmatory results: mapped metrics show strong construct consistency.*** The confirmatory analysis provides strong evidence that PyQu labels are aligned with the QA-specific metric changes that PyQu is designed to capture (Table 1). For **Understandability**, all 7 mapped metrics are significant after BH correction ($q < 0.05$), with the largest separation on Diff scs (Cliff's $\delta = 0.500$, $q = 3.71 \times 10^{-23}$), followed by Diff apifc ($\delta = 0.358$, $q = 6.07 \times 10^{-14}$). For **Reliability**, 4/6 mapped metrics are significant, dominated by Diff 1/d ($\delta = 0.473$, $q = 1.06 \times 10^{-23}$) and Diff scs ($\delta = 0.382$, $q = 1.81 \times 10^{-15}$). For **Maintainability**, 3/4 mapped metrics are significant, with a large effect for Diff LoC ($\delta = -0.481$, $q = 1.61 \times 10^{-15}$), indicating that maintainability-labeled enhancements often coincide with net code reduction. For **Usability**, 6/9 mapped metrics are significant (generally small effects), led again by Diff scs ($\delta = 0.322$, $q = 1.83 \times 10^{-12}$) and Diff 1/cc ($\delta = 0.286$, $q = 7.59 \times 10^{-12}$). In contrast, **Modularity** is markedly weaker: only 1/4 mapped metrics is significant (Diff internal_coupling, $\delta = 0.197$, $q = 9.89 \times 10^{-3}$), while the others show negligible effects. This suggests that modularity improvements are either rarer, more project-dependent, or less captured by the available metric deltas.

| QA | $|\mathcal{M}_{QA}|$ | # sig. | Top mapped metric | Cliff's $\delta$ | BH $q$ |
|---|---|---|---|---|---|
| UN | 7 | 7 | Diff scs | 0.500 | $3.71 \times 10^{-23}$ |
| RE | 6 | 4 | Diff 1/d | 0.473 | $1.06 \times 10^{-23}$ |
| MA | 4 | 3 | Diff LoC | −0.481 | $1.61 \times 10^{-15}$ |
| US | 9 | 6 | Diff scs | 0.322 | $1.83 \times 10^{-12}$ |
| MO | 4 | 1 | Diff int_cp | 0.197 | $9.89 \times 10^{-3}$ |

**Table 1: Confirmatory summary using PyQu-mapped metrics. $|\mathcal{M}_{QA}|$ is the number of metrics mapped to each $QA$; #sig. counts mapped metrics significant after BH correction within $\mathcal{M}_{QA}$. Cliff's $\delta$ is reported with pos=Enhanced. UN: Understandability, RE: Reliability, MA: Maintainability, US: Usability, MO: Modularity, int_cp: Internal_Coupling**

***Exploratory results: a small number of additional signals appear.*** The exploratory analysis over all 17 deltas largely agrees with the confirmatory findings: the strongest signals are mostly within the mapped sets. However, we also observe a few non-mapped metrics emerging among the top discriminators in specific attributes (e.g., Diff 1/d appears among the top signals for Understandability and Usability; Diff external_coupling appears for Reliability and Maintainability). We treat these as exploratory findings that may motivate future refinement of the metric-to-attribute mapping. All confirmatory and exploratory result files (including effect sizes and corrected $q$-values) are included in our replication package.

### Key Findings for RQ1

Quality gains in agentic PRs are real but limited: usability is enhanced most often, whereas modularity is rarely enhanced and exhibits the weakest separation in low-level metric deltas.

When PyQu labels a change as Enhanced, the corresponding mapped metric deltas shift in the expected direction with statistically significant and non-trivial effect sizes—most clearly for understandability and reliability—indicating strong construct-consistency between PyQu's labels and observed metric changes. These results suggest that PyQu can capture relative quality shifts in heterogeneous Python projects beyond its original ML training context, though modularity signals remain weaker.

## 3.2 RQ2: Agentic Code Defects and Security issues

Across the 2,528 Python files we analyzed, most changes were neutral, but a noticeable share either introduced new issues or removed existing ones. Using Pylint to capture code-quality issues, 24.17% (611/2528) of files introduced more issues, while 19.94% (504/2528) removed existing issues, the remaining 55.89% (1413/2528) were neutral. In contrast, using Bandit to capture security issues, the vast majority of files were neutral (91.77%) (2320/2528), with only 4.71% (119/2528) introducing new security issues and 3.52% (89/2528) removing existing ones. Overall, Agentic-PRs show noticeable churn in general code-quality issues, whereas security-related issues are comparatively rare and mostly unchanged.

Table 2 lists the most frequent static-analysis issues in agentic-PR files. On the code-quality side, the distribution is dominated by *convention* messages—especially C0301 (line too long), C0116 (missing function docstring), and C0415 (import outside top level) suggesting that agentic refactorings often satisfy functional goals but frequently miss basic style and documentation expectations. Beyond style, we observe recurring *correctness-relevant* signals such as E1131 (unsupported binary operation), E0402 (relative import beyond top level), and E0602 (undefined variable), as well as common warnings tied to initialization and scoping (e.g., W0233, W0212, W0621). For maintainability, *refactor* messages frequently indicate complex interfaces: R0913 and R0917 (too many arguments / positional arguments) are common, alongside control-flow patterns such as R1705 (no-else-return).

On the security side, Bandit issues are dominated by B101 (use of assert, CWE-703), with much smaller counts for B311 (weak pseudo-random generator, CWE-330) and B603 (subprocess call, CWE-78 context). Overall, the security signals we observe are primarily risky coding practices rather than widespread high-severity vulnerabilities.

A complete mapping from Pylint/Bandit identifiers to message descriptions is provided in the replication package [6].

Table 3 reports the most frequent change operations along with their categories and the static-analysis issues they target. Because a single sampled commit may contain multiple operations that contribute to removing one or more lint or security findings, we sample

| Category | Code | Cnt. | Message |
|---|---|---|---|
| Convention | C0301 | 6824 | Line too long |
| | C0116 | 4775 | Missing function docstring |
| | C0415 | 1216 | Import outside toplevel |
| Error | E1131 | 724 | Unsupported binary operation |
| | E0402 | 243 | Relative beyond top level |
| | E0602 | 157 | Undefined variable |
| Warning | W0233 | 2112 | Non parent init called |
| | W0212 | 1628 | Protected access |
| | W0621 | 1372 | Redefined outer name |
| Refactor | R0913 | 942 | Too many arguments |
| | R0917 | 841 | Too many positional arguments |
| | R1705 | 838 | No else return |
| Security | B101 | 491 | Assert used |
| | B311 | 260 | Pseudo-random generators |
| | B603 | 126 | Subprocess without shell=true |

**Table 2: The frequency of the top 3 codes introduced per issue category. Cnt: Count**

at the commit level but report frequencies over the operation types observed within those commits. Below, we define each operation as used in our taxonomy and explain the relationship between the operation and the flagged issue(s), using our category groupings.

*Security Remediation (Bandit).* These operations are triggered by security-oriented issues from Bandit and, in the best cases, mitigate concrete vulnerabilities. Unlike purely stylistic changes, these fixes often require selecting safer APIs or enforcing runtime validation.

- **Remove Assert** and **Replace assert with explicit runtime validation** both relate to B101, which warns against using assert as a runtime safety mechanism because assertions may be removed under optimization. We distinguish these because their implications differ substantially:
  - *Remove Assert* can be beneficial if the assertion was redundant; however, if the assertion encoded a critical precondition, deletion can *weaken* correctness and security.
  - *Replace assert with explicit runtime validation* is the stronger remediation: it preserves the guard by replacing it with explicit checks that remain active in production (e.g., raising ValueError or domain-specific exceptions).
- **Remove /tmp usage** addresses Bandit B108, which flags hard-coded temporary directories (e.g., /tmp) due to risks such as symlink attacks and unsafe file handling on shared systems. In our dataset, this remediation typically takes the form of removing the explicit /tmp argument and delegating temporary-directory selection to the underlying library (i.e., relying on its default or internally managed temporary path).
- **Remove shell=True** addresses B602, which flags invoking subprocesses through the shell (often enabling command injection when inputs are influenced by untrusted data). Migrating to argument lists (e.g., subprocess.run(["cmd", "arg"])) or safer wrappers generally resolves the root risk by avoiding shell interpretation.
- **Replace Hardcoded Value with Variable** corresponds to B105, which flags hard-coded passwords or credential-like strings. This operation is only a *partial* remediation unless the new variable is sourced from a secure configuration

**Table 3: Categorized change operations, frequencies, and their relationship to static-analysis issues (Target). SR: Security Remediation, MR: Maintainability Refactoring, APIC: API Conformance, SD: Style/Documentation, IH: Import Hygiene, S: Suppression**

| Change Operation | Cat. | Freq. | Target |
|---|---|---|---|
| Remove Assert | SR | 16 | B101 |
| Replace assert with explicit runtime validation | SR | 4 | B101 |
| Remove /tmp usage | SR | 3 | B108 |
| Remove shell=True | SR | 2 | B602 |
| Replace Hardcoded Value with Variable | SR | 2 | B105 |
| Replace with safe library | SR | 2 | B403 |
| Idiomatic Refactoring | MR | 5 | R0205, W0612 |
| Replace Deprecated Method | MR | 4 | W4902 |
| Move Method | MR | 10 | – |
| Extract Method | MR | 5 | R0912 |
| Move file | MR | 1 | – |
| Extract Class | MR | 1 | – |
| Replace Repeated Code with Dynamic Execution | MR | 1 | R1705, R0911, R0912 |
| Replace Explicit Parameters with kwargs | MR | 1 | R0913, R0917 |
| Rename variable | MR | 5 | W0621 |
| API Alignment | APIC | 2 | E1120 |
| Reformat code | SD | 8 | C0301, C0303, W0311 |
| Add Docstring | SD | 6 | C0116 |
| Remove unused import | IH | 7 | W0611 |
| Reorder import | IH | 6 | C0413 |
| Use import | IH | 4 | W0611 |
| Suppresses a lint warning | S | 1 | W0718 |
| Remove the TODO comment | S | 2 | W0511 |
| Remove Code | S | 42 | – |

mechanism (environment variables, secret managers, etc.). Therefore, the warning removal should not be interpreted as evidence of secure secret handling without additional context.

- **Replace with safe library** addresses Bandit B403, which flags imports of potentially unsafe modules due to the risk of arbitrary code execution when deserializing untrusted data. In our dataset, this change appears as replacing `pickle` with a safer serialization format and library, such as JSON. In commit [25], switching from `pickle.dump/load` to `json.dump` preserves the intent of persisting structured data while avoiding `pickle`'s unsafe deserialization semantics; the refactored call additionally specifies formatting options (e.g., `indent` and `separators`) to produce stable, human-readable output.

*Maintainability Refactoring.* These operations primarily restructure code to improve modularity, cohesion, and local reasoning, often reducing maintainability smells reported by linters (e.g., complexity, excessive branching).

- **Idiomatic Refactoring** rewrites semantically equivalent code into more idiomatic Python constructs (e.g., removing useless object inheritance, list `extend`). We observe this correlating with R0205 (useless-object-inheritance) and W0612 (unused-variable, typically by removing redundant patterns and unused loop variables as demonstrated in Listing 1. These transformations are generally low-risk and primarily improve readability and linter conformance.
- **Replace Deprecated Method** updates calls to deprecated APIs to their supported alternatives. While this is often motivated by tool warnings, we categorize it as maintainability refactoring because it improves forward compatibility and reduces technical debt. In our data, it addresses W4902 (deprecated-method).
- **Move Method** relocates a function/method to a different module or class to better align responsibilities and improve cohesion. Because this is a structural transformation, it does not inherently eliminate lint or security issues; rather, it often *shifts* where a finding is reported. Accordingly, the disappearance of a warning in the source location may reflect *relocation* to the destination module/class, not remediation of the underlying issue.
- **Extract Method** decomposes a large function into smaller helpers with narrower responsibilities. In our dataset, this frequently targets R0912 (too many branches) by reducing branching within the original method. Nevertheless, a key nuance is that extraction may distribute complexity across helpers: the warning may disappear at the original site (line) even if total decision logic remains similar.
- **Move file** relocates a module to a different directory/package to improve project organization. This operation typically does not remediate a specific static-analysis finding; instead, it changes the file boundary at which issues are reported and can therefore *relocate* warnings.
- **Extract Class** refactors a large or multi-responsibility module/class by introducing a new class and moving a coherent subset of fields and methods into it. The primary goal is to improve cohesion and separation of concerns rather than to directly remediate a specific static-analysis finding. Consequently, any reduction in reported warnings is often an indirect effect of restructuring, and in some cases issues may simply be *relocated* to the newly extracted class or its hosting module.
- **Replace Repeated Code with Dynamic Execution** replaces repeated conditional blocks (e.g., long *if/elif* cascades) with a dispatch mechanism such as a mapping from keys to callables or policies. In the illustrative example [22], the original implementation repeated the pattern *elif obs.get("action") == ...: return await ...* across many branches, which inflated the number of explicit branches and return sites. The refactored version introduces a typed *POLICY_MAP* from action identifiers to handler lambdas, then performs a constant-time lookup (*handler = POLICY_MAP .get(action)*) followed

by a single conditional invocation. This consolidation reduces syntactic branching and return dispersion, and is therefore consistent with the removal of maintainability issues such as R0912 (*too-many-branches*), R0911 (*too-many-return-statements*), and R1705 (*no-else-return*).

- **Replace Explicit Parameters with `kwargs`** reduces wide interfaces by collecting optional or repeated parameters into keyword arguments. This operation directly addresses R0913 (too many arguments) and R0917 (too many positional arguments).
- **Rename variable** resolves naming/scope issues, most commonly W0621 (redefined outer name), by avoiding shadowing. Although this is a small change, it improves maintainability and can prevent subtle bugs caused by confusing name reuse.

**Listing 1: Remove unused variable `W0612`. Commit 838a1fc3 in evandempsey/fp-growth**

```
1  for i in range(frequency):
2      conditional_tree_input.append(path)
3  conditional_tree_input.extend([path] * frequency)
```

*API Conformance.* This category captures changes that prevent runtime failures caused by mismatched interfaces.

- **API Alignment** addresses E1120 (no-value-for-parameter) by updating call sites to match the callee signature. This often arises after refactoring, dependency upgrades, or changes in internal APIs. Unlike style fixes, API alignment directly affects executability: without it, the code risks raising `TypeError` at runtime.

*Style and Documentation Compliance.* These operations improve consistency with documentation and formatting conventions. They typically do not change program semantics, but they reduce friction in reviews and CI pipelines by aligning with established tooling expectations.

- **Add Docstring** resolves C0116 (missing function/method docstring) by documenting intent and usage. While not correctness-critical, this is a common maintainability requirement in production repositories.
- **Reformat code** resolves formatting-related issues such as C0301 (line too long), C0303 (trailing whitespace), and W0311 (bad indentation). These changes are often mechanical but reflect conformance to shared style guidelines and reduce noise in diffs and reviews.

*Import Hygiene.* This category captures import-focused edits that are closely tied to a small set of linter rules and are primarily intended to improve code cleanliness and conformance to project conventions. Although these changes are often semantics-preserving, they can affect runtime behavior in Python when imports have side effects or when delayed imports were used to avoid circular dependencies.

- **Remove unused import** and **Use import** both address W0611 (unused import). The former removes dead dependencies and reduces namespace clutter. The latter makes an import meaningful by adding a reference (e.g., using a

type, constant, or function), suggesting that the code was incomplete or that the import was intentionally retained.

- **Reorder import** resolves C0413 by moving imports to the recommended location (typically the top of the module) and enforcing a consistent import structure. While this is commonly a style fix, it can be semantically relevant if the original code relied on delayed imports to avoid circular dependencies or expensive initialization. Accordingly, we interpret C0413 fixes as primarily style-driven but potentially requiring deeper restructuring when circularity exists.

*Warning Suppression.* This category highlights changes that reduce tool-reported issues without necessarily addressing the underlying concern. These operations are especially important to separate in analysis, because they can inflate the apparent "quality improvement" if one equates warning disappearance with genuine remediation.

- **Suppresses a lint warning** addresses W0718 by adding an explicit suppression (e.g., `# pylint: disable=...`) rather than changing behavior. This can be justified in boundary layers (interop, CLI wrappers), but it reduces static-analysis coverage and should not be counted as a functional fix.
- **Remove the TODO comment** resolves W0511 by deleting a TODO/FIXME note. This removes the warning but does not necessarily remove the technical debt being tracked; thus, it is best interpreted as tool-output reduction rather than remediation.
- **Remove Code** deletes logic and can eliminate warnings incidentally. However, without contextual evidence (tests, rationale), deletion can also remove safeguards or intended functionality. We therefore treat this operation as semantically ambiguous: it may represent dead-code cleanup, but it may also represent a shortcut to eliminate issues.

---

**Key Findings for RQ2**

Pylint deltas change frequently, and the resulting issues are dominated by style, documentation, and import-discipline violations. In contrast, Bandit issues are comparatively rare and skew toward discouraging *risky idioms* rather than indicating widespread high-severity vulnerabilities. Moreover, the change-operation analysis suggests that apparent "improvements" in static-analysis output should be interpreted cautiously. Several observed operations reduce warnings by *relocating* code (e.g., moving methods/files or extracting classes) or by *removing/suppressing* flagged constructs, rather than by directly remediating the underlying concern. At the same time, we also observe a set of mechanically effective remediations—such as removing `shell=True`, eliminating hard-coded temporary paths, and replacing unsafe serialization libraries—that directly address specific Bandit/Pylint issues.

---

## 3.3 RQ3: Agentic-PRs Status Analysis

Of the 438 Python refactoring PRs we studied, 322 were merged (73.5%), while 84 were closed without merge. Most closed PRs did

not include comments explaining the decision. In a minority of cases, maintainers explicitly stated that the PR was created only to test an agent's capabilities (e.g., [4], "FYI: this was simply a test of Copilot AI. Closing this"), or that similar changes had already landed via other PRs (e.g., [2], "Some of these have been done via separate PRs...Closing this one").

Merge rates also differed across agents. OpenAI Codex accounted for most PRs in the dataset and had the highest observed merge rate, with 262 of 321 PRs merged (81.6%). Cursor PRs were merged in 19 of 31 cases (61.3%), followed by Copilot with 23 of 42 merged PRs (54.8%) and Devin with 16 of 38 merged PRs (42.1%). Claude Code had 2 of 6 PRs merged (33.3%), but this percentage is based on very few instances and should not be overinterpreted. Given the strong imbalance in agent representation, especially the dominance of OpenAI Codex PRs, we treat these agent-level rates as descriptive patterns rather than conclusive evidence of differences in agent quality.

For merged PRs, 239 PRs obtained usable Pylint output. Among these, 46% were merged without introducing new Pylint issues in the modified files. However, 54% (129 PRs) were merged despite introducing at least one new Pylint issue; notably, 73 of these 129 PRs simultaneously removed pre-existing Pylint issues, indicating that improvements and regressions frequently co-occur within the same change set. Remediation practices were heterogeneous: some PRs exhibit an iterative workflow in which authors (or agents) respond to lint regressions with follow-up commits (e.g., [10]), while others were merged without lint-gated checks or reviewer discussion justifying the newly introduced issues.

For security analysis, Bandit reported at least one security finding in 49 merged PRs (15.3% of 322). All 49 removed at least one pre-existing security finding, and 44.9% removed issues without introducing any new Bandit issues. In a qualitative spot-check of PR timelines, we also observed instances where PRs introducing new Bandit issues were merged without visible CI security checks or reviewer discussion (e.g., [23]).

> **Key Findings for RQ3**
>
> Agentic refactoring PRs show high developer acceptance: 73.5% of the studied PRs were merged, and most closed PRs lacked explicit closure rationales. Static-analysis regressions do not necessarily block acceptance: many merged PRs introduced new Pylint issues, often while also removing existing ones, and some PRs introducing Bandit findings were merged without visible security-check discussion.

## 4 Related Works

### 4.1 Quality Analysis and Benchmarks for AI-Generated Code

Evaluation of AI-generated code has traditionally emphasized functional correctness using unit-test benchmarks such as HumanEval [11] and later competitive programming platforms such as LeetCode [5]. However, passing tests does not necessarily reflect maintainability or readability. Recent work therefore adds quality-oriented

analyses that compare AI and human code using static analysis proxies. Ghammam et al. [13] study AI-generated build code in AIDev PRs and find that agentic changes can both introduce and remove maintainability- and security-related build smells. Similarly, Santa Molison et al. [21] use SonarQube rules to analyze Python programming-task solutions and find that LLM-generated code can contain fewer SonarQube bug-type issues and require less estimated effort to fix than human-written solutions. They further report that fine-tuning shifts many high-severity findings toward lower-severity categories, although this comes with a reduction in functional performance. However on harder competition-level tasks, AI-generated solutions can introduce structural critical issues that are uncommon in human-written code, motivating evaluations that consider maintainability, reliability, complexity, and style signals alongside test passing.

### 4.2 Security Analysis of LLM-Generated Code

Another line of work evaluates security risks in AI-assisted and LLM-generated code. Pearce et al.[26] found that Copilot can produce insecure code in security-sensitive scenarios, spanning multiple CWE categories, and Perry et al.[27] showed that developers using AI assistance were more likely to write vulnerable code and often overtrusted the suggestions. Moving to real repositories, Fu et al.[12] report that a substantial fraction of likely AI-written GitHub snippets contain static-analysis security weaknesses. Finally, Yan et al. [33] show that vulnerability-oriented feedback, including self-generated hints and CodeQL reported warnings, can help LLMs avoid or repair insecure code, motivating the integration of static analysis feedback into the generation loop before PR submission.

While prior work motivates evaluating AI-generated code beyond test passing, it provides limited evidence on agent-authored PRs in real workflows. Our study extends this line by analyzing agentic refactoring PRs "in the wild" and jointly characterizing quality-attribute changes (PyQu), lint/security churn (pylint/Bandit), and developer acceptance (merge outcomes).

## 5 Threats to Validity

**Internal threats:** PyQu's prediction models were trained on ML-focused Python projects, which may limit direct generalization to non-ML codebases. We mitigate this threat by verifying construct consistency via a confirmatory analysis that tests whether PyQu labels align with the expected QA-specific metric deltas (Table 1). Moreover, we use AIDev's `refactor` categorization as a practical scoping mechanism to target PRs whose *stated intent* is refactoring. In AIDev, PRs are categorized into one of 11 task types by applying GPT-4.1-mini to the PR title and body, rather than relying solely on repository-side GitHub metadata/labels [18]. This design choice provides a scalable and consistent way to identify refactoring-oriented PRs at dataset scale. Since the categorization is inferred from PR text, some PRs labeled as `refactor` may include mixed changes beyond strictly behavior-preserving refactoring; accordingly, our findings are scoped to PRs categorized as `refactor` in AIDev. Another threat is incorrectly labeling issues by Pylint or Bandit as "new" when warnings shift due to refactoring (e.g., line numbers change or file paths are reformatted). To address this, we normalize file paths and match findings using identifiers such as code/test_id,

message text, and relevant line content rather than relying on raw line numbers. We also run both the parent and child versions with the same settings to avoid differences caused by configuration drift. **External threats:** We use the AIDev dataset [18], which contains AI-generated PRs on GitHub and reflects the specific agents, projects, and workflows captured in that benchmark. As a result, our observations may not generalize to repositories outside this setting (e.g., industrial projects) or to other AI tools not represented in AIDev.

## 6 Ethical considerations

We analyze publicly available GitHub PRs from the AIDev dataset [18] and report only aggregate results, without attempting to identify or link contributors across repositories. Since "agent-generated" labels may be imperfect and could affect perceptions of developers, we avoid attributing intent and focus on measurable signals (quality deltas, static-analysis findings, and code change).

## 7 Conclusion

Overall, agent-generated Python refactoring PRs demonstrate that quality gains are possible but not guaranteed: improvements occur in a minority of changes across the five studied QAs, with usability improving most often. The static-analysis signals are mixed—Pylint issues frequently move in both directions, whereas Bandit issues are less common and change less often. At the code-change level, our manual analysis highlights many lightweight, tool-facing operations (style/documentation compliance, import hygiene, and idiomatic rewrites), complemented by a smaller set of security-oriented remediations and structural refactorings. Importantly, reductions in tool-reported issues do not always imply direct remediation, as issues may also disappear due to code relocation, deletion, or explicit suppression. Despite this variability, developer acceptance is high: 73.5% of the refactoring PRs were merged, and most closed-but-not-merged PRs provided no explicit rationale. Among merged PRs with usable analysis artifacts, new Pylint issues and, less often, new Bandit issues—still appear, often alongside removals of other issues, suggesting that static-analysis regressions do not necessarily block acceptance, especially when changes also remove existing findings. Taken together, agentic refactoring can be beneficial, but it would likely benefit from stronger tool-in-the-loop guardrails that combine quality deltas with lint and security gating prior to PR submission. Finally, our study focuses on characterizing agentic Python refactoring PRs rather than directly comparing them with human-authored refactoring activity. As future work, we plan to construct a matched baseline of human-authored refactoring PRs from the same repositories and comparable time periods. This would enable a more controlled assessment of how agentic refactorings differ from human-authored ones in terms of acceptance, quality impact, and static-analysis outcomes, while reducing confounding effects due to repository context and temporal variation.

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
