# OpenReview forum: "Quality and Security Signals in AI-Generated Python Refactoring Pull Requests"
_ACM.org/AIWare/2026/Conference — AIware 2026_

### Official Review · Reviewer_R2D8 · 2026-03-08

**Rating:** 3
**Confidence:** 3

**Review:**

This paper studies an important and timely question for the community: what kinds of software quality and security signals appear in AI-agent-generated pull requests that are presented as refactorings, and how these signals relate to downstream developer acceptance. This is a meaningful direction because much recent work on AI coding assistants focuses primarily on functional correctness or benchmark success, whereas this paper looks at non-functional software engineering properties in real repository workflows.

I found the paper generally clear, well organized, and easy to follow. The research questions are well scoped, and the study design is coherent. The decision to focus specifically on refactoring PRs is also well motivated. Compared with a broader mix of feature, bug-fix, and maintenance tasks, refactoring is a more controlled setting for examining quality and security effects, since the intended changes are at least nominally behavior preserving.

Strengths:
- **Multi-tool, multi-perspective analysis framework**: The use of PyQu, Pylint, and Bandit gives the study three complementary views of the same artifacts: software quality attributes, lint and code-quality issues, and security warnings. This triangulation is more informative than relying on a single metric or tool.
- **Useful taxonomy of recurring change operations**: The taxonomy of 24 recurring operations is a practical contribution. It helps move the paper beyond aggregate percentages and gives readers a more concrete sense of what AI agents are actually doing when they introduce or remove warnings.
- **Practical acceptance result**: The finding that 73.5% of these PRs are merged, including PRs that introduce new lint or security warnings, is important. It suggests that in real workflows, acceptance is not tightly aligned with static-analysis cleanliness, and that developers may prioritize functionality, speed, or perceived low risk over strict code-quality or security gating.

Weaknesses:
- **Lack of a human-authored baseline**: This is the paper’s biggest limitation. Without a human-authored baseline from the same or similar repositories, it is difficult to know how to interpret the reported rates. For example, it is unclear whether a 22.5% enhancement rate is impressive or weak, whether a 24.17% Pylint introduction rate is unusually high, or whether the 4.7% Bandit introduction rate is concerning relative to normal refactoring practice. The paper gives useful descriptive measurements, but not enough context to judge whether agentic refactoring is better, worse, or roughly comparable to human refactoring.
- **RQ3 is interesting but shallow**: The merge-rate analysis is relevant, but mostly descriptive. The paper reports that many PRs are merged despite introducing warnings, yet does not go much further in explaining what predicts acceptance. For example, it would have been valuable to analyze whether merge likelihood correlates with quality improvements, lint/security regressions, PR size, repository characteristics, or agent identity. Even a simple regression or stratified analysis would have made RQ3 substantially stronger.
- **Some overstatements**: Pylint and Bandit are useful signals, but their outputs do not always map cleanly to practically meaningful regressions. This is especially true for Pylint, where many introduced findings appear to be convention- or style-oriented warnings rather than strong evidence of substantive defects. More importantly, these tools are highly sensitive to project-specific configuration and repository norms. Without stronger evidence that the analysis setup matches each repository’s actual linting or security practices, some of the reported “introduced issues” may overstate the real practical impact.

**Summary:**

This paper presents an empirical study of AI-agent-generated Python refactoring pull requests drawn from the AIDev dataset. The authors analyze 438 refactoring PRs comprising 870 commits produced by five AI agents: OpenAI Codex, Devin, Copilot, Cursor, and Claude Code. To study the effects of these changes, they combine three complementary analysis tools: PyQu for software quality attributes, Pylint for code-quality issues, and Bandit for security issues.

The paper addresses three research questions. For RQ1, it studies whether agentic refactoring improves software quality and finds that 22.5% of commits improve at least one quality attribute, with usability improved most often (36.5%). Statistical testing confirms that PyQu's enhancement labels align with expected metric shifts, particularly for understandability and reliability, though modularity signals remain weak. For RQ2, it measures how often such PRs introduce or remove code-quality and security issues, finding that 24.17% of modified files introduce new Pylint issues, predominantly convention-level violations, and 4.7% introduce new Bandit findings, which are mainly risky coding practices rather than high-severity vulnerabilities. From a manual analysis of high-impact issue removals, the authors derive a taxonomy of 24 recurring change operations and map them to the Pylint/Bandit findings they most commonly affect. Notably, the analysis reveals that apparent improvements in static-analysis output should be interpreted cautiously, as some operations reduce warnings through code relocation, deletion, or suppression rather than direct remediation. For RQ3, it examines developer acceptance and reports a 73.5% merge rate, including for PRs that introduce new lint or security warnings, suggesting that reviewers evaluate changes primarily under a functionality-first lens and tolerate incremental regressions when the perceived net benefit is positive.

---

> ### Author Response · Authors · 2026-03-17
>
> We thank the reviewer for their constructive feedback.
>
> **Human-authored baseline.** We agree that the lack of a matched human-authored refactoring baseline is an important limitation and affects how strongly one can interpret rates such as the 22.5% enhancement rate, the 24.17% Pylint-introduction rate, and the 4.7% Bandit-introduction rate. The current paper is intentionally scoped as an empirical characterization of agentic Python refactoring PRs in the wild, rather than a direct human-vs-agent comparison. We aim to compare agentic-authored refactoring with human-authored refactoring PRs from the same repositories and time window, establishing an important direction for future work with some experiment controls to ensure a fair and meaningful comparison.
>
> **RQ3 depth.** In the current version, RQ3 reports that 73.5% of the 438 refactoring PRs were merged and shows that many merged PRs still introduced new Pylint or Bandit issues, which we interpret as evidence of a functionality-first review lens. We agree that a deeper analysis of acceptance outcomes, such as stratifying by agent type, repository characteristics, PR size, or quality/security deltas, would further strengthen this part of the study. In the camera-ready version, we can extend the analysis by examining the relationship between agent type, PR size, and merge outcome, as noted in our response to Reviewer aa47. The remaining directions would require additional analysis beyond the scope of the current paper, so we will position them more explicitly as important avenues for future work while strengthening the discussion of the current RQ3 findings in the camera-ready version.
>
> **Potential overstatement of introduced issues.** We agree that Pylint and Bandit findings should be interpreted as static-analysis signals, not as perfect proxies for practically meaningful regressions in every repository context. The paper already notes that the most frequently introduced Pylint findings include both convention-level messages and more correctness-/maintainability-relevant warnings, and that the observed Bandit findings are primarily risky coding practices rather than widespread high-severity vulnerabilities. In the camera-ready version, we will strengthen the threats to validity discussion to acknowledge that repository-specific linting configurations may affect the practical weight of these findings.

---

### Official Review · Reviewer_78TY · 2026-03-10

**Rating:** 3
**Confidence:** 4

**Review:**

## Strengths

- Important and timely topic
- Clear multi-tool measurement framework

## Weaknesses

- Lack of a human refactoring baseline
- Limited manual validation

## Major Comments
1. The absence of a human-authored baseline makes several reported metrics difficult to interpret. For example, the reported 24.17% rate of files introducing new Pylint issues and the 73.5% PR merge rate cannot be meaningfully assessed without comparison. The paper should compare these results with human-authored refactoring PRs from the same repositories and time periods.

2. Bandit is a rule-based tool that may produce false positives. For example, B101 (use of assert) is common and acceptable in test code. If the refactoring PRs include changes to test files, some of the reported findings may not represent genuine security risks. The authors should clarify whether the flagged cases are manually validated or filtered to account for such scenarios.

3. The manual validation focuses on 100 commits sampled only from the Not Enhanced group. While this approach can reveal false negatives, it does not verify potential false positives among the Enhanced commits. A more balanced validation strategy that also samples from the Enhanced group would strengthen the conclusions.

4. The paper states that purely stylistic messages are excluded, yet the most frequently introduced Pylint issues include line-length violations and missing docstrings. The authors should clearly define which categories of messages are excluded and explain why the remaining convention-level warnings are treated as quality signals.

5. Table 3 reports frequencies that appear inconsistent with the statement that 100 sampled cases (70 for Pylint and 30 for Bandit). If multiple operations can be assigned to a single case, this should be explicitly stated. The paper should also clarify the unit of analysis (commit or operation) and report the percentages.

**Summary:**

This paper evaluates the effectiveness of agentic refactoring pull requests (PRs) through an empirical study of 438 Python PRs and 870 commits generated by 4 agents from the AIDev dataset. The authors leverage a multi-tool measurement framework consisting of PyQu (assessing five high-level quality attributes), Pylint (identifying general code-quality issues), and Bandit (detecting security-related issues). The results show that coding agents improve at least one quality attribute in 22.5% of cases, with usability showing the highest improvement rate (36.5%) and modularity the lowest (9.5%). In addition, 24.17% of modified files introduce new Pylint issues (mostly stylistic), while 4.7% introduce new Bandit findings, such as risky assert statements. However, these agentic refactoring PRs have a merge rate of 73.5%, implying that developers prioritize structural and functional improvement over strict adherence to automated tool warnings.

---

> ### Author Response · Authors · 2026-03-17
>
> We thank the reviewer for their constructive feedback.
>
> **Human baseline.** We agree that the absence of a matched human-authored refactoring baseline limits contextual interpretation of metrics such as the 24.17% rate of files introducing new Pylint issues and the 73.5% PR merge rate. The current paper is scoped as an empirical characterization of agentic Python refactoring PRs rather than a direct human-vs-agent comparison. We aim to compare agentic-authored refactoring PRs against human-authored refactoring PRs from the same repositories and time periods as an important direction for future work with some experiment controls to ensure fair and meaningful analysis.
>
> **Bandit false positives.** In the current study, we do not manually validate or filter all flagged Bandit introduction cases for context (e.g., B101 in test code). Instead, we use Bandit as a consistent static-analysis signal across commits, while our manual analysis is scoped to a focused sample of issue removals. We agree that this can overstate practically meaningful security risks in some cases, and we will clarify this limitation more explicitly in the threats to validity section of the camera-ready version.
>
> **Manual validation scope.** In the current paper, we manually analyze a random subset of 100 commits from the Not Enhanced group to better understand why PyQu does not flag them. We agree with the reviewer that adding validation from the Enhanced group would strengthen the study by also checking potential false positives, not only false negatives. To partially mitigate this, we complemented the manual analysis with confirmatory and exploratory statistical analyses that test whether the low-level metric shifts align with PyQu’s enhancement labels. These analyses provide additional evidence that the labels are meaningfully associated with expected metric movements, although we agree they do not replace balanced manual validation.
>
> **Stylistic messages and quality signals.** Our intent was to exclude purely cosmetic stylistic messages, such as whitespace or naming-related issues, following prior work, as well as import-unfound warnings. We agree that the current wording may be too broad and could make it unclear why messages such as line-length violations and missing docstrings remain in the analysis. Our rationale is that these warnings, while convention-level, can still affect higher-level code qualities relevant to this study, especially understandability. For example, excessively long lines can reduce readability and make code harder to inspect within a standard screen view, while missing docstrings can hinder understanding of class/function purpose, inputs, outputs, and usage. In the camera-ready version, we will clarify this distinction more explicitly by separating purely cosmetic stylistic exclusions from convention-level warnings retained as lightweight quality signals.
>
> **Table 3 frequencies and unit of analysis.** The total number of operations reported in Table 3 exceeds 100 because a single sampled commit may contain multiple operations that contribute to the removal of one or more lint or security issues. Thus, while the unit of sampling is the commit, the table reports the frequency of observed operation types within those sampled commits. We agree that this distinction should be made more explicit. In the camera-ready version, we will clarify the unit of analysis and state clearly that multiple operations may co-occur within a single commit to eliminate different lint or security issues.

---

> > ### Comment · Reviewer_78TY · 2026-03-18
> >
> > Thank you for the detailed responses. I have a few thoughts regarding the proposed updates for the camera-ready version:
> >
> > **Bandit false positives.** While I appreciate the plan to mention the inclusion of test-code asserts (B101) in the Threats to Validity, I encourage the authors to go a step further by filtering out test files for this rule in the camera-ready version. Identifying test directories and files is generally a straightforward programmatic task. Removing these known false positives (or reporting the proportion) would further strengthen the paper.
> >
> > I thank the authors again for addressing my questions and for providing a clear roadmap for improving the manuscript. I will maintain my score, with the expectation that the promised clarifications and limitations, as well as ideally the test-file filtering for Bandit, will be incorporated into the final camera-ready version.

---

### Official Review · Reviewer_aa47 · 2026-03-11

**Rating:** 4
**Confidence:** 3

**Review:**

## Strengths
* well-written
* uses frontier agentic harnesses
* well-structured replication package with convenient tables for result exploration

## Potential weaknesses
* The dataset is heavily skewed: 321 of 438 PRs come from OpenAI Codex, whereas Claude Code has only 6 PRs
* Filtering of refactoring-labeled PRs depends entirely on the AIDev labeling being accurate, with no validation that these PRs are actually behavior-preserving
* The merge rate analysis (73.5% accepted despite the introduced issue) is the paper's most practically interesting finding but feels underdeveloped. It would be of practical interest  to also check whether merged PRs with regressions correlate with specific agents, repo activity level, or whether follow-up commits clean things up
* Lack of comparison to human-authored refactoring PRs on the same metric framework makes it hard to understand if the 22.5% enhancement rate or 24% Pylint regression rate is better, worse, or typical for refactoring changes in general

**Summary:**

The authors ask whether agentic refactoring PRs from industry-standard coding agents - Codex, Copilot, Devin, Cursor, and Claude Code - actually improve real Python codebases, studying 438 Python PRs across popular GitHub repos filtered from the AIDev dataset.

They apply several complementary tools per commit to identify:
* code quality issues
  * PyQu, an ML classifier trained to identify quality-enhancing commits in Python ML Systems, labeling each commit across five quality attributes (understandability, reliability, maintainability, usability, modularity)
  * Pylint (file-level)
* security issues: Bandit to track issue churn at the file level
* merge outcomes: PR status

---

> ### Author Response · Authors · 2026-03-17
>
> We thank the reviewer for their constructive feedback.
>
> **Dataset skew.** We agree that the dataset is imbalanced across agents: among the 438 refactoring PRs, 321 come from Codex, compared with 42 from Copilot, 38 from Devin, 31 from Cursor, and 6 from Claude Code. This reflects the composition of the filtered AIDev Python-refactoring subset rather than a balanced experimental setup.  In the current paper, we report both aggregate and per-agent results where appropriate, make raw sample sizes explicit, and already note that Claude Code’s percentages may not be informative due to the very small sample. In the camera-ready version, we will clarify that the per-agent results are descriptive and should be interpreted cautiously.
>
> **Refactor label validity.** We use AIDev’s refactor categorization as a practical scoping mechanism to target PRs whose stated intent is refactoring. The AIDev dataset authors automatically classify each PR’s title and body into one of 11 Conventional-Commits-style task categories using GPT-4.1-mini, rather than relying solely on repository-side GitHub metadata/labels [1].  We agree, however, that this methodology does not guarantee every selected PR is a strictly behavior-preserving refactoring. In the camera-ready version, we will clarify this distinction more explicitly in the threats to validity section and state that our findings apply to PRs categorized as refactor in AIDev, rather than to a manually verified set of pure refactorings.
>
> **Merge analysis.** We agree that a deeper investigation into whether merged regressions are associated with specific agents, repository activity, or later cleanup commits would provide valuable additional insight. In the camera-ready version, we can extend the current analysis by examining the relationship between agent type, PR size, and PR merge outcome. However, analyses of post-merge cleanup would require an additional layer of longitudinal investigation beyond the scope of the present study. We will therefore position that aspect more explicitly as an important direction for future work.
>
> **Human baseline.** We agree that a comparison with human-authored refactoring PRs would strengthen the context of our findings. The current paper is intentionally scoped as an empirical characterization of agentic Python refactoring PRs, rather than a direct human-versus-agent comparison. We will position a comparison against human-authored refactoring PRs from the same repositories and similar time periods as an important direction for future work in the camera-ready version, with suitable controls to ensure fairness and meaningful interpretation.
>
>
> [1] HaoLi,HaoxiangZhang,andAhmedE.Hassan.2025. The Rise of AI Team mates in Software Engineering (SE) 3.0 : How Autonomous Coding Agents Are Reshaping Software Engineering. arXiv preprint arXiv:2507.15003(2025).

---

> > ### Comment · Reviewer_aa47 · 2026-03-20
> >
> > Thank you for the detailed response and the specific plan to address the concerns. As my original rating was already `Accept', I will maintain my score.